# Calcium supplementation in pregnancy: An analysis of potential determinants in an under-resourced setting

Atem Bethel Ajong[1,2⊕]*, Bruno Kenfack[3‡], Innocent Mbulli Ali[2‡], Martin Ndinakie Yakum[4‡], Prince Onydinma Ukaogo[5‡], Fulbert Nkwele Mangala[6,7‡], Loai Aljerf[8‡], Phelix Bruno Telefo[2⊕]*

1 Department of Mother and Child care, Kekem District Hospital, Kekem, West Region, Cameroon,
2 Department of Biochemistry, University of Dschang, Dschang, West Region, Cameroon, 3 Department of Obstetrics / Gynaecology and Maternal Health, Faculty of Medicine and Pharmaceutical Sciences, University of Dschang, Dschang, West Region, Cameroon, 4 Department of Epidemiology and Biostatistics, School of Medical and Health Sciences, Kesmonds International University, Bamenda, Cameroon, 5 Department of Pure and Industrial Chemistry, Abia State University, Uturu, Nigeria, 6 Faculty of Medicine and Pharmaceutical Sciences, University of Douala, Douala, Cameroon, 7 Maternity unit, Nkongsamba Regional Hospital, Nkongsamba, Littoral Region, Cameroon, 8 Faculty of Dentistry, Damascus University, Damascus, Syria

⊕ These authors contributed equally to this work.
‡ BK, IMA, MNY, POU, FNM and LA also contributed equally to this work.
* christrah@yahoo.fr (ABA); bphelix@yahoo.co.uk (PBT)

This is a Registered Report and may have an associated publication; please check the article page on the journal site for any related articles.

## Abstract

### Introduction

Despite the evidence that calcium supplementation in pregnancy improves maternofoetal outcomes, many women still do not take calcium supplements during pregnancy in Cameroon. This study identifies factors that influence calcium supplementation during pregnancy in a low resource setting.

### Methods

We conducted a cross-sectional hospital-based study (from November 2020 to September 2021) targeting 1074 healthy women in late pregnancy at the maternities of four major health facilities in the Nkongsamba Health District, Cameroon. Data were collected using an interview-administered semi-structured questionnaire and analysed using Epi Info version 7.2.4.0, and the statistical threshold for significance set at p-value = 0.05.

### Results

The mean age of the participants was 28.20±6.08 years, with a range of 15–47 years. The proportion of women who reported taking any calcium supplements in pregnancy was 72.62 [69.85–75.22]%. Only 12% of calcium-supplemented women took calcium supplements throughout pregnancy, while a majority (50%) took calcium supplements just for 4–5 months. Women believe that taking calcium supplements is more for foetal growth and development (37.12%) and prevention of cramps (38.86%), than for the prevention of

**Data Availability Statement:** All relevant data are within the paper and its Supporting information file.

**Funding:** The author(s) received no specific funding for this work.

**Competing interests:** The authors have declared that no competing interests exist.

hypertensive diseases in pregnancy (2.84%). About all pregnant women (97.65%) took iron and folic acid supplements during pregnancy, and 99.24% took these supplements at least once every two days. Upon control for multiple confounders, the onset of antenatal care before 4 months of pregnancy (AOR = 2.64 [1.84–3.78], p-value = 0.000), having had more than 3 antenatal care visits (AOR = 6.01 [3.84–9.34], p-value = 0.000) and support/reminder from a partner on the necessity to take supplements in pregnancy (AOR = 2.00 [1.34–2.99], p-value = 0.001) were significantly associated with higher odds of taking any calcium supplements in pregnancy.

## Conclusion

Calcium supplementation practices in pregnancy remain poor in this population and far from WHO recommendations. Early initiation of antenatal care, a high number of antenatal visits and reminders or support from the partner on supplement intake significantly increase the odds of taking any calcium supplements in pregnancy. In line with WHO recommendations, women of childbearing age should be sensitised to initiate antenatal care earlier and attain as many visits as possible. Male involvement in prenatal care might also boost the likelihood of these women taking calcium supplements.

## Introduction

As per multiple guidelines, the daily recommended calcium intake varies between 900-1300mg for people 19 years and above. In 2011, about 3.5 billion people were at risk of calcium deficiency, with over 90% found in Africa and Asia [1]. Low mean daily calcium intake (650mg/day) has been reported in Low and Middle-Income Countries (LMIC), and up to 88% of these LMIC had their average calcium intake below 800mg/day [1, 2].

The benefits of calcium supplementation in pregnancy can no longer be overemphasised. According to evidence from multiple systematic reviews with meta-analysis, calcium supplementation in pregnancy significantly reduces the risk of hypertensive diseases in pregnancy, particularly preeclampsia and eclampsia [3–6]. Hypertensive disorders in pregnancy constitute the second leading cause of maternal mortality in Cameroon, contributing to 21–25% of maternal deaths [7, 8].

In addition to having a significant effect on hypertensive diseases in pregnancy, calcium supplementation has other positive effects. It reduces admissions to neonatal high care, pre-term delivery, maternal and neonatal mortality [1, 5, 9], and increases the birth weight [3]. Therefore, low calcium intake in pregnancy is associated with adverse maternofoetal outcomes capable of slowing down the attainment of targets 3.1 and 3.2 of the third sustainable development goal, which focuses on significantly reducing maternal, neonatal and child mortality by 2030 [10].

The above mentioned low calcium intake has been associated with a high prevalence of hypocalcaemia in pregnancy, especially in LMIC [11–14]. In India [15] and Algeria [14], the prevalence of total hypocalcaemia in late pregnancy was found to be as high as 66% and 70%, respectively. Recent evidence in Cameroon suggests that the prevalence of total albumin-corrected hypocalcaemia among women in late pregnancy is around 59% [12].

According to evidence from systematic reviews, LMIC are characterised by suboptimal calcium intake. Women in LMIC are likely to start their pregnancy with already low or low-normal serum calcium levels [2, 16]. Calcium intake in their diet is likely to be insufficient in

meeting their calcium demands, especially in pregnancy [17, 18]. Therefore, calcium supplementation is an indispensable alternative by which these needs can be met.

According to the 2018 guidelines on antenatal care in Cameroon, pregnant women should have at least 8 antenatal care visits during their pregnancy. However, according to the 2018 Demographic and Health Survey (DHS) in Cameroon, 13% of pregnant women did not received medical care from qualified health personnel during their pregnancy. Among the women who received care from qualified health personnel, 65% had attended at least 4 antenatal care visits [19].

Iron and Folic Acid (IFA) remains the only micronutrient combination which is promoted by the national guidelines and all health personnel are trained on its systematic prescription in pregnancy. In pregnancy, calcium and multiple micronutrient supplementations are not recommended in the national guidelines. All supplements, including IFA, are bought by the client or the family [20]. Calcium supplements are prescribed by the health personnel, sometimes on client request, and no standard form of calcium supplements is recommended. To the best of our knowledge, no nationwide programs have been built to promote calcium supplementation in pregnancy. However, some health personnel do not strictly adhere to national guidelines and prescribe calcium supplements to clients based on their experience and client symptoms. Studies measuring the dietary intake of calcium in Cameroon are still lacking.

To prevent hypertensive diseases in pregnancy, the World Health Organisation (WHO) recommends systematic calcium supplementation among pregnant women, especially in countries with low dietary calcium intake [18]. Despite this recommendation, the proportion of women who report calcium supplementation in pregnancy remains relatively low. In China, the proportion of women who took calcium supplements in pregnancy was 57%, and only 11% of these women were adherent to calcium supplements [21]. In a recent survey in Cameroon, up to 43% of pregnant women went through their pregnancy without taking any calcium supplements [12].

Support and reminder from partners and household members, maternal knowledge [21], early antenatal care onset, and a high number of prenatal visits [21, 22] have been reported to affect adherence to calcium and micronutrient supplementation among pregnant women in China [21] and Bangladesh [22]. Other factors include sociodemographic factors like level of education, income, and rural-urban residence [21]. In Africa, particularly in Kenya and Ethiopia, adherence to calcium supplements has been reported to be influenced by on social support [23, 24], and food insecurity/household hunger [24].

Feasibility studies on how to integrate calcium and IFA supplementation in pregnancy to prevent preeclampsia [25] and how to improve the uptake of calcium and IFA through partner adherence support have been carried out in Kenya [23] and Ethiopia [24]. A trial in Kenya has also been conducted to compare a simplified and low-dose antenatal calcium supplementation regimen to the WHO regimen [26], concluding that supplementation with low-dose regimens leads to significantly lower calcium intake. Conclusions from these studies agree that calcium supplementation is feasible in Africa if the supplements are made available for free to the participants and partner/social support methods adopted.

In Africa, particularly Cameroon, studies on calcium supplementation in pregnancy and its determinants are still sparse. Factors that influence calcium supplementation among pregnant women have not yet been studied in Cameroon. The recent study in the Nkongsamba Regional Hospital (Cameroon) evaluated the proportion of women who took calcium supplements (of any form, dose and duration) in pregnancy but did not explore the factors influencing this practice [12]. Knowledge of these factors can contribute significantly to the fight against hypocalcaemia-related adverse outcomes among these women. This study identifies the

sociodemographic and pregnancy-related factors associated with calcium supplementation in pregnancy among women of the Nkongsamba Health District (NHD).

# Materials and methods

## Study design and setting

The methods used to acquire this data have been described and published in a protocol in PLoS One [11]. The procedures related to data acquisition for the above-stated objective are described below. A hospital-based cross-sectional study was conducted from November 2020 to September 2021, targeting pregnant women in the late third trimester. A pretested, semi-structured interview-administered questionnaire was used to collect data, and all analyses were done on the statistical software Epi-Info version 7.2.4.0. Data were collected from four major health facilities of the Nkongsamba Health District (Nkongsamba Regional Hospital, Catholic Medicalised Health Centre, Fultang polyclinic and the Bon Samaritain Medicalised Health Centre).

Nkongsamba is found in the Moungo division, Littoral Region of Cameroon. The population of Nkongsamba town is cosmopolitan, with a total of about 104,050 inhabitants registered from the 2005 population census [11]. Contrary to the information on the original protocol that mentioned the Nkongsamba Regional Hospital (NRH) as the only site for data collection, the three other health facilities were included for a better representation of the NHD, given that they contribute to about 85% of all deliveries in the NHD. The NRH is a third category health facility and serves as the major referral health facility of the NHD and most of the Moungo division [11]. Fultang polyclinic is a fourth category health facility, while the Catholic Medicalised Health Centre and the Bon Samaritain Health Centre are fifth category health facilities. All the health facilities included in this study are located within Nkongsamba urban city. With the findings that reported about 43% of pregnant women going through pregnancy without taking any calcium supplements [11], the NHD was selected to better understand the factors associated with these poor practices in pregnancy.

## Study population, sampling and sample size

Eligible participants were apparently healthy pregnant women residing in Nkongsamba who were received for antenatal care at these four selected health facilities (at least at 37 weeks of gestation). Sampling was exhaustive of all eligible pregnant women, and the minimum required sample size was estimated from the single proportion (Cochran's) formula. For objective number 2 in the registered protocol, we used a proportion of calcium supplementation during pregnancy in the NRH of 57% [12] with precision on either side of 0.03. This was substituted into the formula to give us a minimum required sample size of 1047 participants. As stated in the protocol, different sample sizes were calculated for the different objectives, and the largest sample size (1067) was considered for the study [11].

## Procedure of implementation and data collection

An interviewer-administered semi-structured questionnaire was developed and pretested at the Kekem District Hospital. Administrative authorisations were obtained from the District Medical Officer of the NHD and the directors of the different health facilities. The questionnaires was drafted in English and then translated into French by a team of experts, while the interview was carried out both in French and English depending on the preference of the client. Data collection was done by seven midwives trained in the consenting and data collection procedures.

Upon identifying eligible participants, they were presented the study with a complete informed consent document. For participants who accepted to participate, informed consent forms were signed, and the questionnaire was administered face-to-face and on a one-to-one basis. As reported in the protocol, the questionnaire contained questions to evaluate calcium supplementation practices and the potential factors which were suspected to influence them. This questionnaire was published with the protocol and can easily be accessed [11]. All filled questionnaires were verified for completeness and coherence and then handed to the principal investigator every month.

Calcium supplementation was measured by interview on the practice during pregnancy. In this study, any woman who declared to have taken calcium supplements in any form, posology and for any duration was declared to have taken calcium supplements in pregnancy. The proportion of pregnant women who took calcium supplements was estimated as a fraction of all the women interviewed. The daily dose of calcium taken and the duration of calcium supplementation were calculated from the interview (client's declarations) and information gathered from the medical booklet of the client. The daily dose was estimated from the number of pills taken each day and each pill's elemental calcium content.

## Data analysis

Data from all validated questionnaires were entered into a predesigned data entry sheet made from the statistical software Epi-Info version 7.2.4.0. The amount of missing data was limited in the design and data collection procedure. Our study was cross-sectional, allowing all data to be collected in a single visit. The participant information sheet was developed, consent was always obtained, the questionnaire was client-friendly and easy to use. All data collectors were trained and equipped to limit missing data. Missing data in this study were considered missing completely at random (MCAR). Therefore, the missing data at the end of data collection was managed by the listwise or case deletion technique [27]. In analysing the data, proportions and their 95% confidence intervals (CI) were estimated for categorical variables like calcium supplementation, marital status, and religion. For quantitative variables like age, the mean and its standard deviation were used. For this study, potential predictors of calcium supplementation included age of the participant, her level of education, level of education of partner, household size, gestation age at first visit, number of daily meals, occupation, number of antenatal care visits and support from partner. These factors were selected because they were reported in previous studies to be associated with some preventive interventions' uptake (principally micronutrient and calcium supplementation in pregnancy) [21–24]. This study wanted to check the context specificities of this potential associations.

The strength of association between taking any calcium supplements in pregnancy (dependent variable) and selected covariates like monthly revenue, level of education, and number of antenatal visits was measured using the odds ratio (OR) and its 95% CI by logistic regression. Potential factors with p-values less than 0.25 were included in the multiple logistic regression model, where each variable was controlled for the effect of the others with a threshold of statistical significance at $p = 0.05$.

## Ethics statement

Ethical clearance for this study was obtained from the Cameroon Bioethics Initiative (CAMBIN), Ethics Review and Consultancy Committee (ERCC). The ethical clearance reference number is CBI/452/ ERCC/CAMBIN. Only participants who gave their written and signed consent (assent for minors) were allowed to participate in the study. Participants were free to

leave the study at any time without any penalisation, and their confidentiality and autonomy were respected.

## Results

### Sociodemographic and obstetric characteristics of the participants

This study included a total of 1074 pregnant women in late pregnancy. However, we contacted 1150 eligible participants, and 76 denied participating (non-response rate of 6.61%). The average age of the included participants was 28.20±6.08 years, with a range of 15–47 years. The proportion of women who reported to have taken any calcium supplement in pregnancy stratified by the sociodemographic characteristics of the participants is presented in Table 1.

**Table 1. Characteristics of participants and the proportion of women who took any calcium supplements in pregnancy.**

| Characteristic | Modalities | Frequency (proportion in %) | Proportion of women who took any calcium supplements in pregnancy (%) |
|---|---|---|---|
| Age groups (*n* = 1074) | 15–20 years | 97 (09.03) | 65.98 |
| | 21–30 years | 607 (56.52) | 72.95 |
| | 31–50 years | 370 (34.45) | 73.83 |
| Marital status (*n* = 1068) | Single | 371 (34.74) | 70.17 |
| | Married | 269 (25.19) | 75.19 |
| | Cohabiting | 462 (39.89) | 73.29 |
| | Widow | 02 (0.19) | 50.00 |
| Level of education (*n* = 1074) | Primary | 72 (06.60) | 67.14 |
| | Secondary | 741 (68.99) | 72.81 |
| | Higher | 261 (24.30) | 74.32 |
| Individual monthly income in thousand FCFA (*n* = 1059) | Less than 50 | 119 (11.24) | 70.59 |
| | 50–100 | 270 (25.50) | 73.03 |
| | 100–200 | 456 (43.06) | 70.16 |
| | Above 200 | 214 (20.21) | 77.73 |
| Religion (*n* = 1056) | Atheist | 64 (06.06) | 68.75 |
| | Catholic | 518 (49.05) | 72.90 |
| | Muslim | 33 (03.13) | 63.64 |
| | Protestant | 441 (41.76) | 73.55 |
| Number of people in the house (*n* = 1053) | 1–4 | 438 (41.60) | 76.57 |
| | 5–7 | 415 (39.41) | 71.95 |
| | >7 | 200 (18.99) | 64.14 |
| Number of prenatal visits (*n* = 1034) | No antenatal care | 09 (0.87) | 100.0 |
| | 1–3 | 124 (11.99) | 32.26 |
| | 4–6 | 554 (53.58) | 73.53 |
| | >6 | 347 (33.56) | 87.57 |
| Total number of pregnancies (gravidity) (*n* = 1071) | First pregnancy | 256 (23.90) | 67.59 |
| | 2–4 | 504 (47.06) | 75.75 |
| | 5–8 | 259 (24.18) | 71.28 |
| | >8 | 22 (02.05) | 77.27 |
| Gestational age at first antenatal visit (*n* = 1050) | <2 months | 122 (11.62) | 89.17 |
| | 2–3 months | 263 (25.05) | 85.06 |
| | 4–6 months | 562 (53.52) | 71.61 |
| | ≥7 months | 103 (09.81) | 33.33 |

The sample was dominated by pregnant women aged between 21–30 years (56.52%). A majority (93%) had attended secondary school, and 65.09% were in a union (25.19% legally married and 39.89% cohabiting). A good proportion (87.14%) had attended at least 3 antenatal care consultations, while 63.33% started antenatal care only from 4 months and above.

## Calcium supplementation in pregnancy

The proportion of women who took any calcium supplements during pregnancy was 72.62 [69.85–75.22]%. Fig 1 shows duration of calcium supplementation among calcium supplemented women. Only 12% of the participants took calcium supplements all through pregnancy (from when they knew they were pregnant), while a majority (50%) took calcium supplements only for 4–5 months. Fig 2 shows the daily dose of elemental calcium taken by women who took calcium supplements. Less than half of the calcium supplemented women (47%) took at least 1000mg of elemental calcium per day.

From Table 1, the proportion of pregnant women who took any calcium supplements during pregnancy was found to be highest among women aged 31–50 years (73.83%), among women in a union (74.24%), among participants with higher education (74.32%), among women who earned more than 200.000FCFA (358 USD) a month (77.73%), among women with less than 5 occupants in the house (76.57%), and women who started antenatal care visits within the first two months of pregnancy (85.06%). Concerning the number of prenatal care visits, only 9 women had no antenatal care, and all reported to have taken calcium supplements (100%). Also, 87.57% of women who had more than 6 antenatal visits took calcium supplements.

Table 2 presents information on the context of calcium supplementation. More than 90% of women consulted by health personnel were advised to take calcium supplements during pregnancy. However, women believe that taking calcium supplements is more for foetal growth and development (37.12%) and prevention of cramps (38.86%) than for the prevention of hypertensive diseases in pregnancy (2.84%). Less than 3% of the pregnant women reported

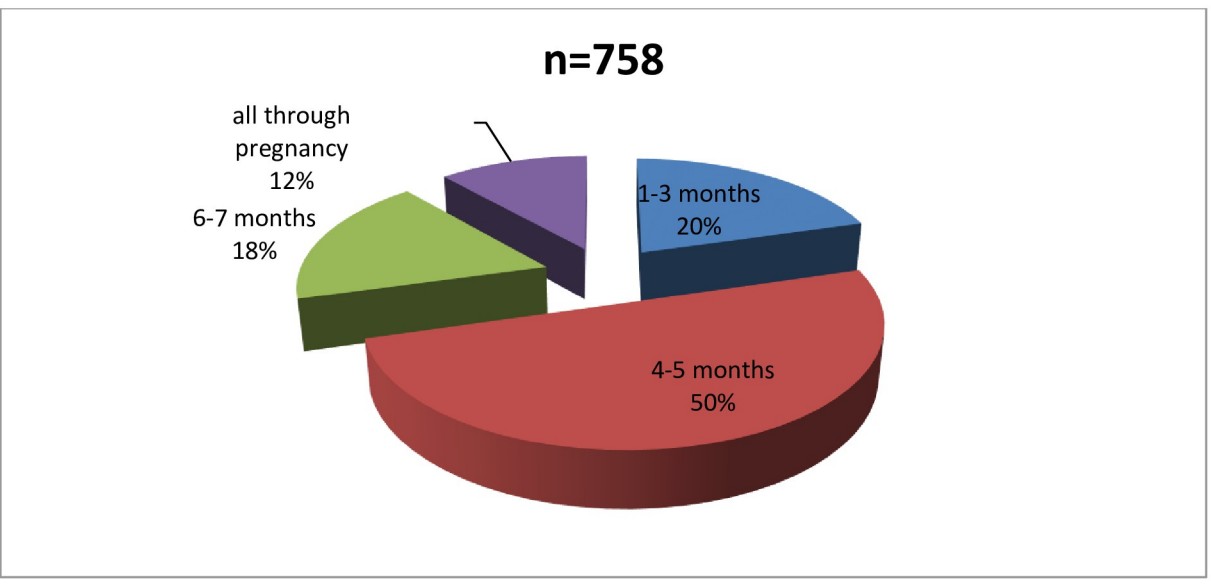

**Fig 1. Duration of taking any calcium supplement among calcium supplemented women.**

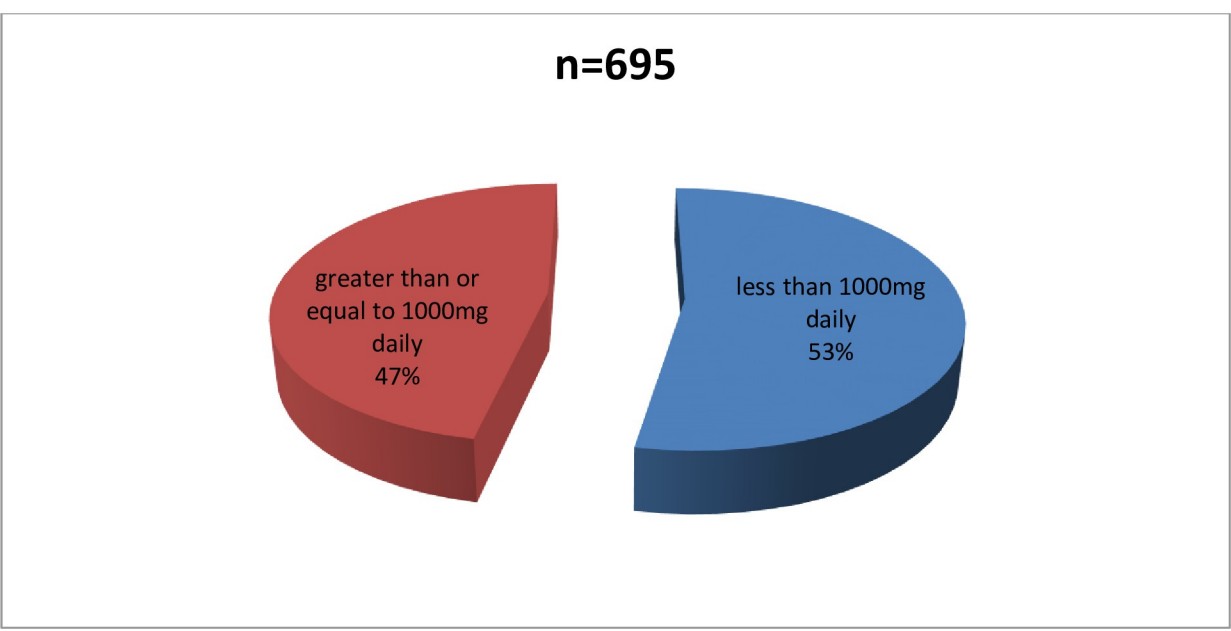

**Fig 2. Dose of calcium intake among calcium supplemented women.**

Table 2. Calcium supplementation and related variables of context.

| Question or variable | Response or modality | Frequency (percentage) |
|---|---|---|
| Personnel adviced her on the necessity to take calcium supplements (n = 856) | Yes | 780 (91.12) |
| | No | 76 (08.88) |
| Importance of taking calcium supplements in pregnancy (n = 916) | For foetal growth and development | 340 (37.12) |
| | Prevention of cramps | 356 (38.86) |
| | Prevention of hypertension | 26 (2.84) |
| | I don't know | 191 (20.85) |
| | Others | 03 (0.33) |
| Did you take iron and folic acid supplements in pregnancy? (n = 1063) | Yes | 1038 (97.65) |
| | No | 20 (1.88) |
| | Took only folic acid supplements | 03 (0.28) |
| | Took only iron supplements | 02(0.19) |
| How often did you take iron and folic acid supplements? (n = 1053) | Every day | 729 (69.23) |
| | Once every two day | 316 (30.01) |
| | Rarely | 08 (0.76) |
| Do you take iron and folic acid supplements at the same time with calcium supplements? (n = 1055) | Yes | 565 (53.55) |
| | No | 204 (19.34) |
| | I don't take calcium supplements | 286 (27.11) |
| Does your partner support you and remind you on the need to take your supplements? (n = 1054) | Always | 284 (26.94) |
| | Sometimes | 602 (57.12) |
| | Occasionally | 120 (11.39) |
| | Never | 48 (04.55) |

having gone through pregnancy without any IFA supplements. Among women who took IFA supplements, 69.23% took them every day, while 30.01% took these supplements once every two days.

## Factors associated with calcium supplementation

Table 3 shows the different factors affecting calcium supplementation in pregnancy, analysed in simple and multiple logistic regression. Following simple logistic regression, a low number of household occupants (<7), earlier onset of antenatal care (before 4 months of pregnancy), a high number of antenatal visits (>3), taking more than 2 meals a day, and reminder or support from partner to take supplements were significantly associated with increased odds of taking any calcium supplements in pregnancy.

However, upon control for confounders in a multiple logistic regression model, the number of meals a day and the number of people in the household disappeared as significant factors. The gestational age at booking antenatal visits, the number of antenatal visits and reminders or support from the partner on the necessity to take supplements were independently and significantly associated with increased odds of taking any calcium supplements in pregnancy. Pregnant women who started antenatal care before 4 months of pregnancy had their odds of taking any calcium supplements during pregnancy significantly increased by 2.64 folds compared to their counterparts who started later (AOR = 2.64 [1.84–3.78], p-value = 0.000). Pregnant women who had more than 3 antenatal visits had their odds of taking any calcium supplements during pregnancy significantly increased by 6 folds compared to those with a lesser number of visits (AOR = 6.01 [3.84–9.34], p-value = 0.000). Also, pregnant women who received support or reminder (sometimes or always) from their partners on the necessity to take supplements had their odds of taking any calcium supplements during pregnancy significantly increased by 2 folds compared to their counterparts with no support or reminder (AOR = 2.00 [1.34–2.99], p-value = 0.001). Factors like level of education, religion, monthly revenue and the number of household occupants had no statistically significant associations with taking any calcium supplements in pregnancy.

**Table 3. Sociodemographic and pregnancy-related factors associated with taking any calcium supplements in pregnancy.**

| Factor | Simple logistic regression | | Multiple logistic regression | |
|---|---|---|---|---|
| | OR [95% CI] | p-value | Adjusted OR [95% CI] | p-value |
| Age below 30 (Y/N) | 0.91 [0.68–1.21] | 0.523 | | |
| Level of education of participant above secondary (Y/N) | 1.12 [0.81–1.54] | 0.482 | | |
| Level of education of partner above secondary (Y/N) | 1.30 [0.96–1.76] | 0.095 [2] | 0.89 [0.63–1.26] | 0.506 |
| Less than 7 household occupants (Y/N) | 1.51 [1.13–2.02] | 0.005 [1,2] | 1.29 [0.92–1.79] | 0.139 |
| Gestational age at first visit below 4 months (Y/N) | 3.30 [2.37–4.61] | 0.000 [1,2] | 2.64 [1.84–3.78] | 0.000 [1] |
| Daily number of meals above 2 (Y/N) | 1.49 [1.13–1.96] | 0.005 [1,2] | 1.38 [1.00–1.90] | 0.053 |
| Number of ANC visits above 3 (Y/N) | 8.55 [5.70–12.83] | 0.000 [1,2] | 6.01 [3.84–9.34] | 0.000 [1] |
| Occupation other than housewife (Y/N) | 1.03 [0.77–1.37] | 0.843 | | |
| Monthly revenue below 100.000 FCFA (179 USD) (Y/N) | 0.99 [0.74–1.30] | 0.918 | | |
| Non-Catholics (Y/N) | 0.97 [0.74–1.28] | 0.841 | | |
| Always or sometimes reminded or supported by partner to take supplements (Y/N) | 2.34 [1.66–3.30] | 0.000 [1,2] | 2.00 [1.34–2.99] | 0.001 [1] |

[1]Statistical significant (p-value less than 0.05)

[2]Eligible for multiple logistic regression (p-value less than 0.25)

Calcium supplementation (Yes/No): Any woman who declared to have taken calcium supplements of any form, for any duration and posology during pregnancy.

## Discussion

The benefits of calcium supplementation in pregnancy can no longer be overemphasised. As stated above, it plays a key role in reducing maternal and neonatal morbi-mortality by directly reducing the likelihood of developing pre-eclampsia and eclampsia. Even though not a recommended practice in national guidelines, the WHO recommends calcium supplementation during pregnancy in Countries like Cameroon. In our study context, women resorted to taking calcium supplements during their pregnancies for some reasons. Contrary to the main goal of the WHO which recommends calcium supplementation for prevention of hypertensive diseases in pregnancy, women took calcium supplements with the belief that it will help in foetal growth and development, and prevent cramps. Only 2.84% of these women could state that it helps in preventing hypertensive diseases in pregnancy. This study was conducted to identify factors associated with calcium supplementation in pregnancy. The proportion of women who reported taking any calcium supplements in pregnancy was 72.62 [69.85–75.22]%. Only 12% of the participants took calcium supplements throughout pregnancy (from when they knew they were pregnant), while a majority (50%) took calcium supplements just for 4–5 months. Less than half of the calcium supplemented women (47%) took at least 1000mg of elemental calcium per day. Upon control for multiple confounders, onset of antenatal care before 4 months of pregnancy (AOR = 2.64 [1.84–3.78], p-value = 0.000), having had more than 3 antenatal care visits (AOR = 6.01 [3.84–9.34], p-value = 0.000) and support/reminder from partner on the necessity to take supplements (AOR = 2.00 [1.34–2.99], p-value = 0.001) were significantly associated with higher odds of taking any calcium supplements in pregnancy. Factors like level of education, religion, monthly revenue and the number of household occupants had no statistically significant associations with taking any calcium supplements in pregnancy.

In our study, 72.62 [69.85–75.22]% of participants reported taking any calcium supplements during pregnancy. This means that 27% went through pregnancy without any form of calcium supplementation. This finding is relatively higher than the previous finding in the NRH (2 years ago), which reported a calcium supplementation proportion of 57.63[52.28–62.80]% [12]. Apart from possible differences linked to sample variation, prescription practices are likely to have changed over the years. According to information from the hospital, about 35% of the personnel in the NRH maternity had been changed a year ago.

Moreover, this study included three other major health facilities in the NHD that could have settings for antenatal care, which encourage better attitudes towards calcium supplementation. The WHO recommends systematic calcium supplementation throughout pregnancy with doses of 1.5–2 g of calcium a day (2–3 times a day) in populations with low dietary calcium intake [18]. This study found significant discordance with this recommendation. Only 12% of calcium supplemented women in this study reported supplementation throughout pregnancy, while up to 20% took calcium supplements for less than 4 months in pregnancy. A majority of women (50%) took supplements for 4–5 months in pregnancy, a duration which is likely to be insufficient in meeting calcium demands. These findings of a short duration of calcium supplementation concord with the previous research results in this setting, which reported a mean supplementation duration of only 4 months [12]. In the same light, relatively low proportions of pregnant women on calcium supplementation have been reported in China (57%) with a meagre adherence rate (11%). Better adherence rates have been reported in studies carried out in Kenya with the use of adherence partner support [23, 27]. Studies in Kenya have also found that adherence rates can be made high with a consistent supply of supplements, health worker training, and family engagement. In a feasibility study where supplements were given free of charge, calcium supplementation adherence rates were as high as 83% [25].

Only 47% of the calcium supplemented women took at least 1000mg of elemental calcium daily during supplementation, while the rest took less than 1000mg. This presents the gaps between recommendations and practice for a zone with low dietary calcium intake and a high prevalence of hypocalcaemia [12]. This considerable discordance between recommendations and practices is likely to significantly increase maternofoetal morbi-mortality associated with preeclampsia and eclampsia [5]. This indirectly affects the attainment of sustainable development targets 3.1 and 3.2 [10].

When each of the 6 eligible factors from simple logistic regression was controlled in a multiple logistic (each one controlled for the effect of the other 5), three factors revealed statistically significant associations with taking any calcium supplements in pregnancy. The factor with the highest strength of association was the number of antenatal visits.

Women who had more than 3 antenatal visits had their odds of taking any calcium supplements during pregnancy significantly increased by 6 folds compared with their counterparts with a lesser number of antenatal visits (p-value = 0.000). A higher number of antenatal visits has been associated with better obstetric outcomes [28, 29]. The number of antenatal care visits is likely to have increased the probability of better antenatal health education [30] and probable prescription of calcium by health personnel. Also, early antenatal visits (before 4 months) significantly increased the odds of taking any calcium supplements in pregnancy by 2.64 folds. Early initiation of antenatal care has independently been associated with better obstetric outcomes [31, 32]. Similar findings were reported in a recent study in Bangladesh, where early onset and a high number of antenatal visits were significantly associated with good supplementation practices [22]. Moreover, a large scale cross-sectional study in China also reported quality antenatal care and the number of visits to be significantly associated with adherence to micronutrient supplementation in pregnancy [21]. These findings align with the current WHO recommendations on antenatal care (adopted by Cameroon), which recommend the first antenatal contact before 12 weeks and at least 8 antenatal care visits during pregnancy for a positive pregnancy experience [17].

Moreover, partner reminders and support on calcium supplementation doubled the odds of calcium supplementation in pregnancy. The Bangladeshi study [22] reported the same relationship in their study in 2017. This brings out the necessity for male involvement in antenatal care. In a recent systematic review and meta-analysis, male involvement in antenatal care was associated with skilled birth attendance utilization [33]. Male involvement in antenatal care could improve the likelihood of adherence to supplementation in pregnancy. Concurring findings have been reported in Kenya and Ethiopia [24].

Our study did not evaluate the stock availability of calcium, the prescription attitudes of physicians, or the purchasing power/acceptability of calcium supplements which are potential reasons for not supplementing. However, 91.12% of our participants reported that they were advised by the health personnel to take calcium supplements, while 89.69% declared that they supplemented with calcium prescribed by the health personnel. Therefore, prescription by health personnel is at the centre as early antenatal care onset or number of antenatal care visits might increase the likelihood of having a calcium supplement prescription.

Contrary to findings in the large scale Chinese study conducted in 2019 [21], the level of education and income had no statistically significant association with taking any calcium supplements in pregnancy. This could be explained by the enormous sample size of the study (30,027 women), which gave it a very high power [34]. Therefore, the effect attributed to these factors is likely to be very small [35]. Also, a factor like the number of household occupants reported by Nguyen *et al* in 2017 appeared to be a confounder during multiple logistic regression [22]. The daily number of meals analysed in simple logistic regression also appeared as a confounder in the multiple logistic regression model.

The results herein should be interpreted with care. The cross-sectional design of this study makes it impossible to establish cause-effect relationships between selected variables and calcium supplementation. This research did not consider potential factors affecting adherence to calcium supplementation like perceived side effects (gastric disturbances and constipation), support from the community, and other groups (parents, children, or other family members). Moreover, potential reasons for non-supplementation like stock availability of calcium, the prescription attitudes of physicians, the purchasing power/acceptability of calcium supplements, and previous counseling about calcium supplementation were not evaluated. Calcium supplementation was not measured using an adherence scale. Any woman who declared to have taken calcium supplements during pregnancy (of any dose, duration and form) was classified to have supplemented with calcium. Therefore, women who took supplements for less than 1 month were included. However, up to 62% of calcium supplemented women took supplements for at least 4 months and 47% of calcium supplemented women took at least 1000g of elemental calcium a day. Despite these limits, this study identifies three independent factors, with a logical and sufficient explanation that independently and significantly affect calcium supplementation among pregnant women in this setting. Moreover, our study was conducted with steps taken to limits different forms of bias which could affect our results. Selection bias was limited by consecutively including every healthy pregnant woman in to the study during the period of data collection. All consenting participants were interviewed alone in closed environments, and some of the information was collected from their medical booklets to limit information or measurement bias. Also, to reduce measurement bias, all data collectors were well trained to ensure adequate measurement or collection of the study variables. Moreover, the principal investigator checked all questionnaires for completeness and coherence.

## Conclusion

Calcium supplementation practices in pregnancy in this population are far from recommended WHO standards. Three main factors significantly and independently influence the likelihood of taking calcium supplements in pregnancy. These factors include early initiation of antenatal care (before 4 months), having more than 3 antenatal visits during pregnancy and partner reminder or support on the necessity of taking supplements in pregnancy. The number of antenatal care visits has the strongest association with calcium supplementation.

Apart from encouraging systematic calcium supplementation in pregnancy and systematically prescribing calcium supplements during antenatal care, the consumption of locally available high calcium-containing foods and fruits should be encouraged. Sensitisation of women of childbearing age and their partners on the importance of early antenatal care initiation, and the need to cover a minimum of 8 antenatal visits as recommended by the WHO are indispensable. Male partner involvement in antenatal care should be encouraged to improve women's adherence to calcium supplementation. This could be done by sending personal invitation cards for male partners, community mobilization, mass media communication, or by home visits.

## Supporting information

**S1 File. Data base of determinants of calcium supplementation in pregnancy.**
(MDB)

## Acknowledgments

Our sincere gratitude goes out to:

- The Director of the NRH and Bethanie Group of laboratories for their support,

- Fouko Eric Dagobert and collaborators for their assistance in data collection,

- Midwives who participated in data collection, and

- Pregnant women who consented to participate in this study.

A number of changes have occurred on the author list compared with the registered protocol published. Additional authors include Prince Onydinma Ukaogo and Fulbert Nkwele Mangala who have contributed significantly by supervision, validation, review and editing.

## Author Contributions

**Conceptualization:** Atem Bethel Ajong, Bruno Kenfack, Innocent Mbulli Ali, Loai Aljerf, Phelix Bruno Telefo.

**Data curation:** Atem Bethel Ajong, Martin Ndinakie Yakum, Loai Aljerf, Phelix Bruno Telefo.

**Formal analysis:** Atem Bethel Ajong, Bruno Kenfack, Martin Ndinakie Yakum, Fulbert Nkwele Mangala.

**Funding acquisition:** Atem Bethel Ajong.

**Investigation:** Atem Bethel Ajong, Bruno Kenfack, Innocent Mbulli Ali, Martin Ndinakie Yakum, Prince Onydinma Ukaogo, Fulbert Nkwele Mangala, Loai Aljerf, Phelix Bruno Telefo.

**Methodology:** Atem Bethel Ajong, Bruno Kenfack, Innocent Mbulli Ali, Martin Ndinakie Yakum, Loai Aljerf, Phelix Bruno Telefo.

**Project administration:** Atem Bethel Ajong, Bruno Kenfack, Innocent Mbulli Ali, Loai Aljerf, Phelix Bruno Telefo.

**Resources:** Atem Bethel Ajong, Prince Onydinma Ukaogo, Fulbert Nkwele Mangala, Loai Aljerf.

**Software:** Atem Bethel Ajong, Martin Ndinakie Yakum, Prince Onydinma Ukaogo.

**Supervision:** Atem Bethel Ajong, Bruno Kenfack, Innocent Mbulli Ali, Loai Aljerf, Phelix Bruno Telefo.

**Validation:** Atem Bethel Ajong, Bruno Kenfack, Innocent Mbulli Ali, Martin Ndinakie Yakum, Prince Onydinma Ukaogo, Fulbert Nkwele Mangala, Loai Aljerf, Phelix Bruno Telefo.

**Visualization:** Atem Bethel Ajong, Prince Onydinma Ukaogo, Fulbert Nkwele Mangala, Phelix Bruno Telefo.

**Writing – original draft:** Atem Bethel Ajong.

**Writing – review & editing:** Atem Bethel Ajong, Bruno Kenfack, Innocent Mbulli Ali, Martin Ndinakie Yakum, Prince Onydinma Ukaogo, Fulbert Nkwele Mangala, Loai Aljerf, Phelix Bruno Telefo.

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
