## [Decision Letter · Decision Letter 0]

23 May 2022

PONE-D-21-31679Calcium supplementation in pregnancy: an analysis of potential determinants in a resource-limited settingPLOS ONE

Dear Dr.  Ajong,

Thank you for submitting your manuscript to PLOS ONE. After careful consideration, we feel that it has merit but does not fully meet PLOS ONE’s publication criteria as it currently stands. Therefore, we invite you to submit a revised version of the manuscript that addresses the points raised during the review process.

Please revise this manuscript point by point as per the review comments. I also suggest the authors should address the research questions presented in the registered report protocol and should support the conclusions by the data. 

We look forward to receiving your revised manuscript.

Kind regards,

Tesfaye Hambisa Mekonnen

Academic Editor

PLOS ONE

Journal Requirements:

3. Please ensure that you refer to Figure 1 and 2 in your text as, if accepted, production will need this reference to link the reader to the figure.

Reviewers' comments:

Reviewer's Responses to Questions

**Comments to the Author**

1. Does the manuscript adhere to the experimental procedures and analyses described in the Registered Report Protocol?

If the manuscript reports any deviations from the planned experimental procedures and analyses, those must be reasonable and adequately justified.

Reviewer #1: Yes

2. If the manuscript reports exploratory analyses or experimental procedures not outlined in the original Registered Report Protocol, are these reasonable, justified and methodologically sound?

A Registered Report may include valid exploratory analyses not previously outlined in the Registered Report Protocol, as long as they are described as such.

Reviewer #1: Yes

3. Are the conclusions supported by the data and do they address the research question presented in the Registered Report Protocol?

The manuscript must describe a technically sound piece of scientific research with data that supports the conclusions. The conclusions must be drawn appropriately based on the research question(s) outlined in the Registered Report Protocol and on the data presented.

Reviewer #1: Partly

4. Have the authors made all data underlying the findings in their manuscript fully available?

Reviewer #1: Yes

5. Is the manuscript presented in an intelligible fashion and written in standard English?

Reviewer #1: Yes

6. Review Comments to the Author

Please use the space provided to explain your answers to the questions above. (Please upload your review as an attachment if it exceeds 20,000 characters)

Reviewer #1: Thank you for the opportunity to review this paper about factors that influence calcium supplementation in Nkongsamba Cameroon. Calcium supplementation has been recommended by WHO for years but there are few countries where it is included as part of ANC and there is still limited research about the factors that influence adherence. This is an important study to help understand more about factors that influence Ca supplementation. However, this paper can be strengthened by providing more information about the context/setting and a clearer definition of adherence measures. My comments are below:

Overall

More information about ANC in Cameroon is needed to provide context for this paper. Is Ca supplementation recommended by the Ministry of Health? Are Ca supplements provided for free as part of ANC? (it sounds like women are purchasing Ca supplements which is surprising, how does this differ from IFA is that provided for free or are women used to purchasing their own supplements). How many supplements are recommended and provided? Is there information about the types of Ca supplements that are provided (mg)? How do women know how many mg they are taking? What are the rates of preeclampsia/hypertensive disorders in the country? How many ANC visits are recommended by the MOH? How many ANC visits do most women make based on the authors' previous, DHS, or other data? What about other micronutrient supplementation programs in the country as part of ANC. Is IFA recommended, multiple micronutrient supplements? Has Ca supplementation been promoted? Have ANC providers been trained on Ca supplementation? Is there data about dietary Ca intake in Cameroon and/or Nkongsamba.

A section on measures needs to be added to the methods. Each calcium supplementation measure needs to be defined and the process or collecting this information and creating categories needs to be described. The use of the word prevalence is unclear as it is used throughout this paper. Noting that the interview guide is available in the protocol paper would help readers who want more detail. But a short summary is also needed.

How do the findings from this study compare with IFA supplement adherence in the area? Does it seem consistent? Do women in Nkongsamba typically purchase IFA supplements in pregnancy?

These data were collected during COVID-19, how did that impact access to ANC, calcium supplies, etc?

In the conclusion, the authors highlight the need to involve male partners. Are other family members such as grandmothers (pregnant women's mothers and mothers-in-law) also influential during this time? Would it potentially be important to involve them as well?

The following paper may be helpful for framing the importance of this research

Gomes F, Ashorn P, Askari S, Belizan JM, Boy E, Cormick G, Dickin KL, Driller‐Colangelo AR, Fawzi W, Hofmeyr GJ, Humphrey J. Calcium supplementation for the prevention of hypertensive disorders of pregnancy: current evidence and programmatic considerations. Annals of the New York Academy of Sciences. 2022 Jan 8.

Specific comments

Title – consider another term than low-resource setting. Could either name the area Nkongsamba or Cameroon or use under-resourced setting

Introduction

Lines 61-68 - I suggest starting the introduction with the second paragraph about the health consequences of low calcium intake to help the reader understand why Ca supplementation is recommended, and then the paragraph with information about prevalence of low calcium in subsequent paragraphs

Line 69 – use of the word refined here is not clear, suggest deleting

Line 71 – While the protocol focuses on serum Ca levels this seems less relevant for this paper

Line 75 – clarify this is pregnant women

Line 82 – is this pregnant women in China, please clarify.

Line 82 – should this be affect calcium supplementation “adherence”?

Line 83 – food insecurity/household hunger has been associated with poorer adherence to Ca supplements in Kenya (see the following article : Martin SL, Omotayo MO, et al. Adherence partners are an acceptable behaviour change strategy to support calcium and iron‐folic acid supplementation among pregnant women in Ethiopia and Kenya. Maternal & Child Nutrition. 2017 Jul;13(3):e12331.)

Line 86-87 – Please refer to papers by Omotayo et al describing a study on Ca supplementation in Kenya

Omotayo MO, Dickin KL, Pelletier DL, Martin SL, Kung'u JK, Stoltzfus RJ. Feasibility of integrating calcium and iron–folate supplementation to prevent preeclampsia and anemia in pregnancy in primary healthcare facilities in Kenya. Maternal & child nutrition. 2018 Feb;14:e12437.

Omotayo MO, Dickin KL, Pelletier DL, Mwanga EO, Kung'u JK, Stoltzfus RJ. A simplified regimen compared with WHO guidelines decreases antenatal calcium supplement intake for prevention of preeclampsia in a cluster-randomized noninferiority trial in rural Kenya. The Journal of nutrition. 2017 Oct 1;147(10):1986-91.

Line 88 - Please include a reference for this study about prevalence?

Line 88 – what is meant by supplementation prevalence? Please clarify if it is pregnant women taking any Ca, taking recommended amount of Ca (and stating what that amount is)? Something else?

Line 90 – is obstetric the correct word here? Is it related to child birth or pregnancy? I believe it is pregnancy.

Line 95 - clarify that this is referring to a protocol

Line 95 – clarify that this is a secondary data analysis

Line 97 and 116 – was it a structured survey or semi-structured interview guide? Typically I consider a semi-structured interview guide to be used to collect qualitative data and it suggests the interviewer has flexibility in terms of which questions are asked and how - I imagine all questions were meant to be asked in the order they were written?

Line 102 - Please provide more information about Nkongsamba are all these health facilities in urban areas? Other than NRH what level hospital are the others?

Line 110 – how was sample size determined?

Line 110 – Please describe attempts that were made to address potential bias.

Line 121 – Please note how many participants refused to participate.

Line 127 – What language were the interviews conducted in?

Line 127 – How was calcium supplementation measured and defined? How was it asked? I see there are descriptions of daily dose, how was this determined?

Line 129 – Please describe how missing data were handled

Table 1 and line 134. Is revenue household income or her individual income? Please clarify

Line 155 – please use married in place of union

Line 158 – having the definition of the Ca measures in the methods would really help this paragraph

Line 161-162, what is meant by all through pregnancy – every day all through pregnancy? Every day since they knew they were pregnant? Is there a recommendation for when calcium supplementation should start in Cameroon? Is it at 20 weeks or before? Is Ca supplementation mentioned in national ANC guidelines?

Table 2 – what is the definition of Ca supplementation in this table?

Line 205 – clarify what is meant by prevalence of calcium supplementation in pregnancy?

Line 225 – specify this is in populations with low dietary calcium intake (not in LMIC)

Line 234 – in Kenya with consistent supply, health worker training, family engagement adherence rates were quite high. Consider referencing these findings as well as the low adherence rates in Kenya

Omotayo MO, Dickin KL, Pelletier DL, Mwanga EO, Kung'u JK, Stoltzfus RJ. A simplified regimen compared with WHO guidelines decreases antenatal calcium supplement intake for prevention of preeclampsia in a cluster-randomized noninferiority trial in rural Kenya. The Journal of nutrition. 2017 Oct 1;147(10):1986-91.

Line 249 – include reference for this statement

Line 250 – include reference for this statement

Line 253 – please clarify what is meant by obstetric outcomes, I read it as maternal and infant mortality/birth outcomes but based on the sentence that follows it seems to be adherence to supplements – please clarify. If it is obstetric outcomes please add a reference, if supplement please clarify.

Line 256 – what is meant by antenatal follow up?

Line 257 is this supplementation adherence?

Line 260 – is this also the recommendation in Cameroon?

Line 266 – Consider referring to this paper which focuses on partner support

Martin SL, Omotayo MO, et al. Adherence partners are an acceptable behaviour change strategy to support calcium and iron‐folic acid supplementation among pregnant women in Ethiopia and Kenya. Maternal & Child Nutrition. 2017 Jul;13(3):e12331.

Line 276 include year for reference

Line 284 – is this referring to perceived side effects?

Line 286 – what about counseling about calcium

Figure 2 – how was daily dose of Ca determined?

Thank you for including the interview guide in the protocol paper. Including the results from some of these questions would help provide context for the results

1. Have you been advised by health personnel on the necessity to take calcium supplements? a)Yes b)No

2. Why is it important to take calcium supplements in pregnancy? a) I don’t know b) For foetal bone development and growth c) to prevent cramps in the mother d) prevent high blood pressure in pregnancy, e)others

3. Did you take iron and folic acid supplements in pregnancy? a) Yes b) No , c)I took only folic acid, e)I took only iron supplements.

4. If yes, How often? a) Everyday b) At least once every two days c) Rarely

5. Do you take calcium supplements at the same time as the iron and or folic acid supplements? a) Yes b)No c) I don’t take calcium supplements

Also it would be important to note that partner support question was not specific to calcium supplements but appears to be asking about both IFA and calcium supplements, please clarify this.

6. Does your partner support you and remind you on the need to take your supplements? a) Always b)Sometimes c)Occasionally d) Never

7. PLOS authors have the option to publish the peer review history of their article (what does this mean?). If published, this will include your full peer review and any attached files.

Reviewer #1: No

---

## [Author Response · Author response to Decision Letter 0]

29 May 2022

Response to reviewer’s comments

Editor’s comments and Responses

Comment: Please revise this manuscript point by point as per the review comments. I also suggest the authors should address the research questions presented in the registered report protocol and should support the conclusions by the data. 

Resp: Thank you for your recommendation. We presented a point by point response to reviewers on this table. This manuscript is presented just for objective 2 of the objectives presented on the registered protocol. Other questions raised on this protocol have been published (1) and others are under review (PONE-D-21-33966R3). We have tried to answer all reviewers’ comments and hope our conclusions are in line with the objectives of this manuscript.

Comment: Please ensure that your manuscript meets PLOS ONE's style requirements, including those for file naming. 

Resp: Our manuscript has been formatted to respect PLOS One guidelines. The files have been named accordingly.

Comment: We note that you have stated that you will provide repository information for your data at acceptance. Should your manuscript be accepted for publication, we will hold it until you provide the relevant accession numbers or DOIs necessary to access your data. If you wish to make changes to your Data Availability statement, please describe these changes in your cover letter and we will update your Data Availability statement to reflect the information you provide. 

Resp: Thanks for suggestion. We have now changed our data availability statement to “All relevant data are within the paper and its Supporting information file”. The data base has been submitted with this revision as a supporting information file.

Comment: Please ensure that you refer to Figure 1 and 2 in your text as, if accepted, production will need this reference to link the reader to the figure. 

Resp: These two figures have been mentioned in the text. See line 258-264.

Reviewer’s comments and Responses

Comment: Thank you for the opportunity to review this paper about factors that influence calcium supplementation in Nkongsamba Cameroon. Calcium supplementation has been recommended by WHO for years but there are few countries where it is included as part of ANC and there is still limited research about the factors that influence adherence. This is an important study to help understand more about factors that influence Ca supplementation. However, this paper can be strengthened by providing more information about the context/setting and a clearer definition of adherence measures. My comments are below: 

Resp: Thank you for your assessment. We greatly appreciate the time and efforts put in to extensively review this manuscript and recommend the changes. We must agree that this has gone a long way in strengthening and bringing more sense to this piece of work. We have addressed each of your comments and provided explanations and amendments to the manuscript. We hope that our efforts meet you expectations.

Comment: Overall

More information about ANC in Cameroon is needed to provide context for this paper. Is Ca supplementation recommended by the Ministry of Health? Are Ca supplements provided for free as part of ANC? (it sounds like women are purchasing Ca supplements which is surprising, how does this differ from IFA is that provided for free or are women used to purchasing their own supplements). How many supplements are recommended and provided? Is there information about the types of Ca supplements that are provided (mg)? How do women know how many mg they are taking? What are the rates of preeclampsia/hypertensive disorders in the country? How many ANC visits are recommended by the MOH? How many ANC visits do most women make based on the authors' previous, DHS, or other data? What about other micronutrient supplementation programs in the country as part of ANC. Is IFA recommended, multiple micronutrient supplements? Has Ca supplementation been promoted? Have ANC providers been trained on Ca supplementation? Is there data about dietary Ca intake in Cameroon and/or Nkongsamba. 

Resp: Thanks for the suggestions. The whole introduction has been reviewed as recommended. Please see the introduction, page 4-7.

 Comment: A section on measures needs to be added to the methods. Each calcium supplementation measure needs to be defined and the process or collecting this information and creating categories needs to be described. The use of the word prevalence is unclear as it is used throughout this paper. Noting that the interview guide is available in the protocol paper would help readers who want more detail. But a short summary is also needed. 

Resp: Thank you for the recommendation. This information has been included. See paragraph 2, page 10.

The interview guide has also been cited in the methods. See line 194, paragraph 1, page 10

Comment: How do the findings from this study compare with IFA supplement adherence in the area? Does it seem consistent? Do women in Nkongsamba typically purchase IFA supplements in pregnancy?

 Resp: We have now presented data on the IFA supplementation on table 2. See last paragraph page 16 and page 17. As stated in the background, all supplements in pregnancy as purchased by the client.

Comment: These data were collected during COVID-19, how did that impact access to ANC, calcium supplies, etc?

Resp: Thanks for the question. We have no study around the hospital which was aimed at evaluating the impact of COVID-19 ANC access. However, we belief that COVID-19 might have reduce the number of ANC contacts for some women. But given the fact our data was collected in late pregnancy, we think that most women even if did not have a complete ANC follow-up, they would have shown up in late pregnancy. All supplements during ANC are prescribed and are bought by the client. We don’t have information or data on how COVID-19 might have affected calcium supplies in the pharmacies.

Comment: In the conclusion, the authors highlight the need to involve male partners. Are other family members such as grandmothers (pregnant women's mothers and mothers-in-law) also influential during this time? Would it potentially be important to involve them as well? 

Resp: Thank you for your question. This has been included as a limit of our study given that we failed to collect data on the support of other groups and family members. See last paragraph of discussion, line 419

Comment: The following paper may be helpful for framing the importance of this research

Gomes F, Ashorn P, Askari S, Belizan JM, Boy E, Cormick G, Dickin KL, Driller‐Colangelo AR, Fawzi W, Hofmeyr GJ, Humphrey J. Calcium supplementation for the prevention of hypertensive disorders of pregnancy: current evidence and programmatic considerations. Annals of the New York Academy of Sciences. 2022 Jan 8. 

Resp: Thank you very much for your extensive research and review. This article was of great help to us. We have added it to our references.

Specific comments reviewer’s comments and Responses

Comment: Title – consider another term than low-resource setting. Could either name the area Nkongsamba or Cameroon or use under-resourced setting 

Resp: Thank you for your suggestion. Under-resourced setting has been used in the title. See title

Introduction

Comment: Lines 61-68 - I suggest starting the introduction with the second paragraph about the health consequences of low calcium intake to help the reader understand why Ca supplementation is recommended, and then the paragraph with information about prevalence of low calcium in subsequent paragraphs

Resp: Thanks for your suggestion. The whole introduction structure has been reviewed and we tried to organise ideas as recommended. See introduction.

Comment: Line 69 – use of the word refined here is not clear, suggest deleting 

Resp: Thanks very much. We have now taken off the word refined. See line 94.

Comment: Line 71 – While the protocol focuses on serum Ca levels this seems less relevant for this paper 

Resp: Thank you for your comment. The statement was just to emphasise the consequence of low calcium intake on calcaemic states from recent reviews, thereby indicating the need for calcium supplementation.

Comment: Line 75 – clarify this is pregnant women 

Resp: Thank you for the vigilance. The word “pregnant” has been added. See line 123.

Comment: Line 82 – is this pregnant women in China, please clarify. Line 82 – should this be affect calcium supplementation “adherence”? 

Resp: Thank you for the question. This is in pregnancy (China and Bangladesh). The word “pregnant” has been added. See line 114. Thanks for the request of specificity. Actually these factors affected adherence. See line 123.

Comment: Line 83 – food insecurity/household hunger has been associated with poorer adherence to Ca supplements in Kenya (see the following article : Martin SL, Omotayo MO, et al. Adherence partners are an acceptable behaviour change strategy to support calcium and iron‐folic acid supplementation among pregnant women in Ethiopia and Kenya. Maternal & Child Nutrition. 2017 Jul;13(3):e12331.) 

Resp: Thanks for the information. This information has been included in the background section. See line 125-127.

Comment: Line 86-87 – Please refer to papers by Omotayo et al describing a study on Ca supplementation in Kenya

Omotayo MO, Dickin KL, Pelletier DL, Martin SL, Kung'u JK, Stoltzfus RJ. Feasibility of integrating calcium and iron–folate supplementation to prevent preeclampsia and anemia in pregnancy in primary healthcare facilities in Kenya. Maternal & child nutrition. 2018 Feb;14:e12437.

Omotayo MO, Dickin KL, Pelletier DL, Mwanga EO, Kung'u JK, Stoltzfus RJ. A simplified regimen compared with WHO guidelines decreases antenatal calcium supplement intake for prevention of preeclampsia in a cluster-randomized noninferiority trial in rural Kenya. The Journal of nutrition. 2017 Oct 1;147(10):1986-91. 

Resp: Thank you very much for these references, they have been used in the background. See line 125-135.

Comment: Line 88 - Please include a reference for this study about prevalence?

Resp: Thank you for your vigilance. The reference has been added. See line 141.

Comment: Line 88 – what is meant by supplementation prevalence? Please clarify if it is pregnant women taking any Ca, taking recommended amount of Ca (and stating what that amount is)? Something else?

Resp: Thank you for the question. The precision has been made throughout the manuscript. The “prevalence” has been taken of. The proportion of women who report to have taken calcium supplements (any form, dose and duration) in pregnancy is used. It has also been defined in the methods. See 197-204.

Comment: Line 90 – is obstetric the correct word here? Is it related to child birth or pregnancy? I believe it is pregnancy.

Resp: Thanks for the suggestion. The word has been changed to “pregnancy-related”. See line 143.

Comment: Line 95 - clarify that this is referring to a protocol

Resp: Thanks very much. This has been made clear. See line 148

Comment: Line 95 – clarify that this is a secondary data analysis

Resp: Thank you for the suggestion. This manuscript was designed to respond to objective 2 of the registered protocol which intended to identify barriers to calcium supplementation among these women.

Comment: Line 97 and 116 – was it a structured survey or semi-structured interview guide? Typically I consider a semi-structured interview guide to be used to collect qualitative data and it suggests the interviewer has flexibility in terms of which questions are asked and how - I imagine all questions were meant to be asked in the order they were written?

Resp: Thanks for your questions; the questions were not always asked the way they were written on the questionnaire. In some cases and some questions, the meaning of the question was further explained to the client to make sure the information given was adapted to the question asked.

Comment: Line 102 - Please provide more information about Nkongsamba are all these health facilities in urban areas? Other than NRH what level hospital are the others? 

Resp: Thank you for the request. The requested information has been provided. See line 157-167.

Comment: Line 110 – how was sample size determined?

Resp: Thank you for your question. Information on the determination of the sample size has been presented. See line 175-180.

Comment: Line 110 – Please describe attempts that were made to address potential bias.

Resp: Thanks very much. A section has been added on bias. See line 205-211.

Comment: Line 121 – Please note how many participants refused to participate.

Resp: Thanks for the request. The non-response rate has been presented in the first paragraph of the result section. See line 239-240.

Comment: Line 127 – What language were the interviews conducted in?

Resp: Thanks for the question. The interview was conducted either in English or French depending on the preference of the client. This is indicated on the manuscript. See line 185-187. 

Comment: Line 127 – How was calcium supplementation measured and defined? How was it asked? I see there are descriptions of daily dose, how was this determined?

Resp: Thank you for the question. This has been presented in the methods. See line 197-204.

Comment: Line 129 – Please describe how missing data were handled 

Resp: Thanks for the recommendation. This information has been added. See line 215-221.

Comment: Table 1 and line 134. Is revenue household income or her individual income? Please clarify 

Resp: Thanks for the question. This referred to her individual monthly income. See table 1.

Comment: Line 155 – please use married in place of union

Resp: Thanks for suggestion. Union is used here to designate legally married women and those who are cohabiting. The precision has been made to the phrase. See line 245-246.

Comment: Line 158 – having the definition of the Ca measures in the methods would really help this paragraph Resp: Thank you for the suggestion. This has been added to the methods section as requested.

Comment: Line 161-162, what is meant by all through pregnancy – every day all through pregnancy? Every day since they knew they were pregnant? Is there a recommendation for when calcium supplementation should start in Cameroon? Is it at 20 weeks or before? Is Ca supplementation mentioned in national ANC guidelines?

Resp: Thanks for request for the precision. All through pregnancy was used for women who declared that they took calcium supplements from the time they knew they were pregnant. This precision has been included in bracket. See line 261.

Cameroon has no recommendation as to when to initiate calcium supplementation in pregnancy and calcium supplementation in pregnancy is not a standard practice as per the national ANC guidelines. 

comment: Table 2 – what is the definition of Ca supplementation in this table?

Resp: Thanks for the question. The definition of calcium supplementation in the study has now been presented in the methods section and table 2. See page 9, last paragraph and table 2.

Comment: Line 205 – clarify what is meant by prevalence of calcium supplementation in pregnancy?

Resp: Thanks for the question. The prevalence of calcium supplementation in pregnancy has now been defined in the methods section. See page 10, line 197-204

Comment: Line 225 – specify this is in populations with low dietary calcium intake (not in LMIC)

Resp: Thank you very much for the correction. This correction has been made. See line 350-51

Comment: Line 234 – in Kenya with consistent supply, health worker training, family engagement adherence rates were quite high. Consider referencing these findings as well as the low adherence rates in Kenya

Omotayo MO, Dickin KL, Pelletier DL, Mwanga EO, Kung'u JK, Stoltzfus RJ. A simplified regimen compared with WHO guidelines decreases antenatal calcium supplement intake for prevention of preeclampsia in a cluster-randomized noninferiority trial in rural Kenya. The Journal of nutrition. 2017 Oct 1;147(10):1986-91.

Resp: Thank you for your suggestion. This information has been included and referenced. See line 360-364.

Comment: Line 249 – include reference for this statement

Line 250 – include reference for this statement 

Resp: Thanks for the recommendation. These statements have been referenced. See line 379-381.

Comment: Line 253 – please clarify what is meant by obstetric outcomes, I read it as maternal and infant mortality/birth outcomes but based on the sentence that follows it seems to be adherence to supplements – please clarify. If it is obstetric outcomes please add a reference, if supplement please clarify.

Resp: Thanks for the suggestion. Its obstetric outcomes and the references have been added. See line 380.

Comment: Line 256 – what is meant by antenatal follow up?

Resp: Thank you for the question. Antenatal “care” has been used in the place of “follow-up”. See line 387.

Comment: Line 257 is this supplementation adherence?

Resp: Thank you for the question. It has to do with adherence. The precision has been made in the manuscript. See line 388.

Comment: Line 260 – is this also the recommendation in Cameroon?

Resp: Thanks for the question. This WHO recommendation is adopted for Cameroon. The precision has been added to the sentence. See line 390.

Comment: Line 266 – Consider referring to this paper which focuses on partner support

Martin SL, Omotayo MO, et al. Adherence partners are an acceptable behaviour change strategy to support calcium and iron‐folic acid supplementation among pregnant women in Ethiopia and Kenya. Maternal & Child Nutrition. 2017 Jul;13(3):e12331. 

Resp: Thank you for the suggestion. This paper has been read and referenced. See line 398.

Comment: Line 276 include year for reference 

Resp: Thanks for the request. The year of the reference has been added. See line 407.

Comment: Line 284 – is this referring to perceived side effects? 

Response: Thanks for the question. We meant perceived side effects. See line 418.

Comment: Line 286 – what about counseling about calcium 

Resp: Thank you for the addition. This has been integrated into a limit. See line 421-422.

comment: Figure 2 – how was daily dose of Ca determined? 

Resp: Thank you for the question. This information has been provided in the methods section.

Comment: Thank you for including the interview guide in the protocol paper. Including the results from some of these questions would help provide context for the results

1. Have you been advised by health personnel on the necessity to take calcium supplements? a)Yes b)No

2. Why is it important to take calcium supplements in pregnancy? a) I don’t know b) For foetal bone development and growth c) to prevent cramps in the mother d) prevent high blood pressure in pregnancy, e)others

3. Did you take iron and folic acid supplements in pregnancy? a) Yes b) No , c)I took only folic acid, e)I took only iron supplements.

4. If yes, How often? a) Everyday b) At least once every two days c) Rarely

5. Do you take calcium supplements at the same time as the iron and or folic acid supplements? a) Yes b)No c) I don’t take calcium supplements

Also it would be important to note that partner support question was not specific to calcium supplements but appears to be asking about both IFA and calcium supplements, please clarify this.

6. Does your partner support you and remind you on the need to take your supplements? a) Always b)Sometimes c)Occasionally d) Never 

Resp: Thank you for your recommendation. We have now included table 2 with analysis of these variables.

References

1. Ajong AB, Kenfack B, Mbulli Ali I, Yakum MN, Ukaogo PO, Mangala FN, et al. Ionised and total hypocalcaemia in pregnancy: An analysis of prevalence and risk factors in a resource-limited setting, Cameroon. Spradley FT, editor. PLoS One [Internet]. 2022 May 18 [cited 2022 May 26];17(5):e0268643. Available from: https://journals.plos.org/plosone/article?id=10.1371/journal.pone.0268643

---

## [Decision Letter · Decision Letter 1]

18 Aug 2022

PONE-D-21-31679R1Calcium supplementation in pregnancy:  an analysis of potential determinants in an under-resourced settingPLOS ONE

Dear Dr. Ajong,

Thank you for submitting your manuscript to PLOS ONE. After careful consideration, we feel that it has merit but does not fully meet PLOS ONE’s publication criteria as it currently stands. Therefore, we invite you to submit a revised version of the manuscript that addresses the points raised during the review process.

We look forward to receiving your revised manuscript.

Kind regards,

Tesfaye Hambisa Mekonnen

Academic Editor

PLOS ONE

Journal Requirements:

Reviewers' comments:

Reviewer's Responses to Questions

**Comments to the Author**

1. Does the manuscript adhere to the experimental procedures and analyses described in the Registered Report Protocol?

If the manuscript reports any deviations from the planned experimental procedures and analyses, those must be reasonable and adequately justified.

Reviewer #1: Yes

2. If the manuscript reports exploratory analyses or experimental procedures not outlined in the original Registered Report Protocol, are these reasonable, justified and methodologically sound?

A Registered Report may include valid exploratory analyses not previously outlined in the Registered Report Protocol, as long as they are described as such.

Reviewer #1: Yes

3. Are the conclusions supported by the data and do they address the research question presented in the Registered Report Protocol?

The manuscript must describe a technically sound piece of scientific research with data that supports the conclusions. The conclusions must be drawn appropriately based on the research question(s) outlined in the Registered Report Protocol and on the data presented.

Reviewer #1: Yes

4. Have the authors made all data underlying the findings in their manuscript fully available?

Reviewer #1: Yes

5. Is the manuscript presented in an intelligible fashion and written in standard English?

Reviewer #1: Yes

6. Review Comments to the Author

Please use the space provided to explain your answers to the questions above. (Please upload your review as an attachment if it exceeds 20,000 characters)

Reviewer #1: Thank you for the opportunity to review this revised manuscript about calcium supplementation in Cameroon. The manuscript has been substantially improved and the information about the contextual information in Cameroon is helpful to understand the setting.

All line numbers refer to the tracked changes version of the manuscript.

It is still surprising that health workers are recommending/prescribing calcium supplements since it is not included in the ANC guidelines. It would be helpful if the authors could provide some explanation for why ANC providers are recommending calcium (Lines 110-1112)

It would be helpful if throughout the manuscript any time the authors reference women have taken calcium supplements or womens calcium supplementation that it would be written as have taken any calcium supplements just to make it clear that it is not referencing a specific amount dose or duration. While this is noted in the methods it would be a helpful reminder to the reader if it were throughout the manuscript.

Line 35 – please revise proportion of women who reported calcium supplementation to proportion of women who reported [taking any] calcium supplements

Line 63-64 please add references

Line 102 – consider deleting only as this is not such a low number (there are several places where only, just, meager, other words are used, consider reviewing and revising whether or not these are appropriate)

Line 105 – please add promoted by who - the MOH or in the national ANC guidelines?

Line 126 consider revising "depend on" to "influenced by"

Line 129 – add "support" to partner adherence [support]

Line 205 the bias section could be moved to the end of the discussion in a limitations paragraph

Line 217, please clarify, what is the study document?

Line 224 please revise "between calcium supplementation" to "taking any calcium supplements" (and change throughout eg lines 315-315, 324 but many other places as well)

Line 257 – please add "any" – took any calcium supplements

Line 274 – consider deleting the word some

Line 279– please add any – without taking any IFA supplements

Line 323-339, rather than restating the results could describe women’s understanding of why to take supplements and provide more of an explanation

Line 340 – please revise, could consider something like: In our study 72.62 [69.85-75.22]% of participants reported taking any calcium supplements during pregnancy.

Line 345 – why are prescription practices likely to have changed?

Line 355 – please add a reference

Line 360-361, mean adherence in these studies was 86%, this is not low, consider revising

Line 412 add date for Nguyen et al reference

Conclusion – should Cameroon adopt the WHO guidelines about calcium supplementation

Line 444 please delete “their “from their women’s adherence

Line 444-445 - do your study findings justify a recommendation about involving men in ANC, if so consider adding this recommendation and references in discussion earlier. There are examples of challenges with involving men in ANC in other settings in Africa and several studies about male involvement in ANC related to PMTCT and not. Are there other ways to involve male partners to support calcium supplementation based on the literature (for example from the study in Bangladesh) that could complement recommendations about involving men in ANC (community mobilization, mass media, home visits) since getting men to ANC visits can be challenging.

7. PLOS authors have the option to publish the peer review history of their article (what does this mean?). If published, this will include your full peer review and any attached files.

Reviewer #1: No

---

## [Author Response · Author response to Decision Letter 1]

22 Aug 2022

Thank you for the opportunity to review this revised manuscript about calcium supplementation in Cameroon. The manuscript has been substantially improved and the information about the contextual information in Cameroon is helpful to understand the setting.

All line numbers refer to the tracked changes version of the manuscript.

 Resp: Thank you very much for taking the time to review our manuscript. We are grateful

It is still surprising that health workers are recommending/prescribing calcium supplements since it is not included in the ANC guidelines. It would be helpful if the authors could provide some explanation for why ANC providers are recommending calcium (Lines 110-1112)

 Resp: Thanks for your observation. Health workers sometimes violate national guidelines especially when growing evidence is showing the benefits of the practice. Some health professionals do prescribe calcium supplements systematically, some prescribe depending on the client’s symptoms and request. Line 103 to 104 has been added to make this clear. 

It would be helpful if throughout the manuscript any time the authors reference women have taken calcium supplements or womens calcium supplementation that it would be written as have taken any calcium supplements just to make it clear that it is not referencing a specific amount dose or duration. While this is noted in the methods it would be a helpful reminder to the reader if it were throughout the manuscript.

 Resp: Thank you for your suggestion. We have edited the manuscript to make this clear following your suggestion. 

Line 35 – please revise proportion of women who reported calcium supplementation to proportion of women who reported [taking any] calcium supplements

Line 63-64 please add references

 Thank you for the suggestion. It has been corrected. See line 37. Also the reference has been added, see line 65

Line 102 – consider deleting only as this is not such a low number (there are several places where only, just, meager, other words are used, consider reviewing and revising whether or not these are appropriate)

 Resp: Thanks for your observation. “Only” was used to say that it is far from target. 87% means 13% went through pregnancy without receiving medical care from a qualified health personnel. See line 92-95. I have also reviewed the manuscript to see if the words used are appropriately placed.

Line 105 – please add promoted by who - the MOH or in the national ANC guidelines?

Line 126 consider revising "depend on" to "influenced by"

 Resp: Thank you for the suggestions. This corrections has been made. See line 96 and 97 and line 119.

Line 129 – add "support" to partner adherence [support]

 Resp: Thank you for the correction. We have addressed this on the manuscript. See line 123.

Line 205 the bias section could be moved to the end of the discussion in a limitations paragraph

Line 217, please clarify, what is the study document?

 Resp: Thanks you for the review. The bias section has also been carried to the end of the discussion as requested. See 421-428. Also, correction has been made to clarify the study document. It is the participant information sheet. See line 209.

Line 224 please revise "between calcium supplementation" to "taking any calcium supplements" (and change throughout eg lines 315-315, 324 but many other places as well)

 Resp: Thank you for this correction. The whole document has been verified accordingly. See line 217.

Line 257 – please add "any" – took any calcium supplements

 Resp: Thanks for the suggestion. This has been added. See line 246.

Line 274 – consider deleting the word some

Line 279– please add any – without taking any IFA supplements

Line 323-339, rather than restating the results could describe women’s understanding of why to take supplements and provide more of an explanation

 Resp: Thanks for the suggestions. These corrections were adopted. See lines 261, 266, and line 305-314. However, we prefer to have a small reminder of the key findings of the study here.

Line 340 – please revise, could consider something like: In our study 72.62 [69.85-75.22]% of participants reported taking any calcium supplements during pregnancy.

 Resp: Thanks for your suggestion. This was adopted. See lines 331-333.

Line 345 – why are prescription practices likely to have changed? 

Resp: Thank you for the question. We think this might have been caused by a change in the personnel of the department which could have affected practices.

Line 355 – please add a reference 

Resp: Thanks for the request. The reference has been added. See line 349.

Line 360-361, mean adherence in these studies was 86%, this is not low, consider revising 

Resp: Thank you for the correction. The sentence has been reformulated. See lines 351-353.

Line 412 add date for Nguyen et al reference 

Resp: Thanks very much. The date has been added. See line 403.

Conclusion – should Cameroon adopt the WHO guidelines about calcium supplementation

 Resp: Thank you for the question. Our study evaluated factors influencing calcium supplementation. We did not re-evaluate the benefits of calcium supplementation in pregnancy. So based only on the results of this manuscript, we cannot give this recommendation. However, substantial evidence recognises calcium supplementation in pregnancy to have beneficial results for the mother and child. We found out that with respect to WHO standards, calcium supplementation in Cameroon in pregnancy is not carried out adequately. We encourage women to supplement in calcium while fighting to meet most of their calcium needs in the diet. However, we think Cameroon should adopt the WHO recommendations on calcium supplementation in pregnancy. 

Line 444 please delete “their “from their women’s adherence 

Resp: Thank you for the suggestion. The word has been deleted. See line 442

Line 444-445 - do your study findings justify a recommendation about involving men in ANC, if so consider adding this recommendation and references in the discussion earlier. There are examples of challenges with involving men in ANC in other settings in Africa and several studies about male involvement in ANC related to PMTCT and not. Are there other ways to involve male partners to support calcium supplementation based on the literature (for example from the study in Bangladesh) that could complement recommendations about involving men in ANC (community mobilization, mass media, home visits) since getting men to ANC visits can be challenging.

 Resp: Thank you for the question. Our study results suggest that the male partner could contribute to the client’s adherence to prescribed protocols. This indirectly means that men could play a role. We had this information added in the discussion section. See lines 384-390. 

Thanks for the question. We have added a few precisions on how men could be involved. See lines 443-444.

---

## [Decision Letter · Decision Letter 2]

12 Oct 2022

PONE-D-21-31679R2Calcium supplementation in pregnancy:  an analysis of potential determinants in an under-resourced settingPLOS ONE

Dear Dr. Ajong,

Thank you for submitting your manuscript to PLOS ONE. After careful consideration, we feel that it has merit but does not fully meet PLOS ONE’s publication criteria as it currently stands. Therefore, we invite you to submit a revised version of the manuscript that addresses the points raised during the review process. Please try to address some of the technical issues Reviewer #2 has raised which I also believe that it should be addressed. 

We look forward to receiving your revised manuscript.

Kind regards,

Tesfaye Hambisa Mekonnen

Academic Editor

PLOS ONE

Journal Requirements:

Reviewers' comments:

Reviewer's Responses to Questions

**Comments to the Author**

1. Does the manuscript adhere to the experimental procedures and analyses described in the Registered Report Protocol?

If the manuscript reports any deviations from the planned experimental procedures and analyses, those must be reasonable and adequately justified.

Reviewer #1: Yes

Reviewer #2: Yes

2. If the manuscript reports exploratory analyses or experimental procedures not outlined in the original Registered Report Protocol, are these reasonable, justified and methodologically sound?

A Registered Report may include valid exploratory analyses not previously outlined in the Registered Report Protocol, as long as they are described as such.

Reviewer #1: Yes

Reviewer #2: Partly

3. Are the conclusions supported by the data and do they address the research question presented in the Registered Report Protocol?

The manuscript must describe a technically sound piece of scientific research with data that supports the conclusions. The conclusions must be drawn appropriately based on the research question(s) outlined in the Registered Report Protocol and on the data presented.

Reviewer #1: Yes

Reviewer #2: Partly

4. Have the authors made all data underlying the findings in their manuscript fully available?

Reviewer #1: Yes

Reviewer #2: Yes

5. Is the manuscript presented in an intelligible fashion and written in standard English?

Reviewer #1: Yes

Reviewer #2: Yes

6. Review Comments to the Author

Please use the space provided to explain your answers to the questions above. (Please upload your review as an attachment if it exceeds 20,000 characters)

Reviewer #1: The authors have comprehensively responded to previous comments. Their revisions have strengthened this manuscript. I do not have any additional comments.

Reviewer #2: It is important study in which authors invesigated the factors affecting supplementantion of calcium. There are some interesting findings. However, some technical issues should be adressed further if a revision is invited.

1. Becasue the aim of this study is to invetstigate the affecting factors, what potential facors did the authors cared about? And why to select them? Authors should clearly state this informaiton in the part of methods.

2. In the part of data anlaysis, author said that "the mean (µ) ...." it is wrong expression. For smaple analysis, µ is not apprioriate because it is for population.

3. For logisitc regression, it is unclear about depentent variable. A clear definition is reqiured. Moreover, if dependent variable is "the proportion of women who took any calcium supplements during pregnancy", logistic model could not be suitable because the propotion is hiher (72.6%), which could overestimate the odds. Possion regeression is suggested.

4. Due to few covariates considered in this study, all covariates are suggested to be included in the adjusted analysis not just included covariates which are significant at P less than 0.25 in simple logisitc regression.

7. PLOS authors have the option to publish the peer review history of their article (what does this mean?). If published, this will include your full peer review and any attached files.

Reviewer #1: No

Reviewer #2: No

---

## [Author Response · Author response to Decision Letter 2]

15 Oct 2022

RESPONSE TO REVIEWERS

We appreciate the time taken by the reviewers and editors to fully evaluate and strengthen this manuscript. We have now responded to the comments raised by the reviewer. This is presented here in a tabular form. As mentioned in the cover letter during the last revision, we also have made some changes on the reference list. The correct version of reference 16 has been provided. Also reference 17 has been taken off (it was a retracted version of reference 2). Also reference 26 and 27 have been interchanged.

Comment Response

1. Becasue the aim of this study is to invetstigate the affecting factors, what potential facors did the authors cared about? And why to select them? Authors should clearly state this informaiton in the part of methods. 

 Response: For this study, potential predictors of calcium supplementation included age of the participant, her level of education, level of education of partner, household size, gestation age at first visit, number of daily meals, occupation, number of ANC and support from partner. These factors were selected because they were reported I previous study to be associated with some preventive interventions’ uptake. The authors wanted to check the context specificities of this potential association. This has been added in the methods. See data analysis section, line 205-211.

2. In the part of data anlaysis, author said that "the mean (µ) ...." it is wrong expression. For smaple analysis, µ is not apprioriate because it is for population. 

 Response: Thank you for this comment. We agree with the reviewer, and we have taken this symbol off. see lines 204, 205 and 228.

3. For logisitc regression, it is unclear about depentent variable. A clear definition is reqiured. Moreover, if dependent variable is "the proportion of women who took any calcium supplements during pregnancy", logistic model could not be suitable because the propotion is hiher (72.6%), which could overestimate the odds. Possion regeression is suggested. 

Response: Thank you for this comment. 

1. We have included clear definition of the outcome variable at the level of methodology of the manuscript. See lines 183-190 and line 212.

2. It is unclear to the authors, the reason the reviewer thinks that higher proportion of outcome variable in the study sample would lead to overestimation of the odds in logistic regression. A more elaborate description on how logistics regression is used can be seen in the book Logistic Regression Models by Joseph M. Helbe (https://doi.org/10.1201/9781420075779) equally, a systematic review (https://www.scirp.org/journal/paperinformation.aspx?paperid=95655° ) describe the section of outcome variable for logistic regression. In all, the decision to use logistic model is based more on the nature of the outcome variable and not on value of the outcome. Moreover, we endeavored to present our associations with precisions that these were “odds” not “risk” nor likelihood. 

3. Besides, Poisson regression can not be applied in this study because it is recommended for studies with count outcomes and not categorical nor continues outcomes (https://www.tandfonline.com/doi/abs/10.1080/01621459.1987.10478502 )

4. We believe that Logistic regression remains the best method to make this analysis based on the dichotomic nature of the outcome variable.

4. Due to few covariates considered in this study, all covariates are suggested to be included in the adjusted analysis not just included covariates which are significant at P less than 0.25 in simple logisitc regression. 

Response: Thank you very much for this comment. We agree with the reviewer that fewer independent variable in the logistic model may make the model to explain greater or lesser variance than it is in reality. However, we would like to cite a section of this review on logistics regression in medical research (https://www.scirp.org/journal/paperinformation.aspx?paperid=95655 ) on this issue. It states “Unfortunately, the solution is not simply to include as many variables as possible, as the inclusion of variables that are unrelated to the outcome in question, this (the addition of unrelated variables) has the tendency to inflate the apparent predictive validity of the final model”. Several methods exist for the selection of the variables to be included in the model (doi: 10.1136/fmch-2019-000262) and the use of 0.25 p-value cutoff from bivariate analysis aims to reduce the inclusion of unrelated variables.

---

## [Decision Letter · Decision Letter 3]

24 May 2023

PONE-D-21-31679R3Calcium supplementation in pregnancy:  an analysis of potential determinants in an under-resourced settingPLOS ONE

Dear Dr. Ajong,

Thank you for submitting your manuscript to PLOS ONE. After careful consideration, we feel that it has merit but does not fully meet PLOS ONE’s publication criteria as it currently stands. Therefore, we invite you to submit a revised version of the manuscript that addresses the points raised during the review process.

We look forward to receiving your revised manuscript.

Kind regards,

Ikechukwu Innocent Mbachu

Academic Editor

PLOS ONE

Journal Requirements:

Reviewers' comments:

Reviewer's Responses to Questions

**Comments to the Author**

1. Does the manuscript adhere to the experimental procedures and analyses described in the Registered Report Protocol?

If the manuscript reports any deviations from the planned experimental procedures and analyses, those must be reasonable and adequately justified.

Reviewer #1: Yes

Reviewer #2: Yes

2. If the manuscript reports exploratory analyses or experimental procedures not outlined in the original Registered Report Protocol, are these reasonable, justified and methodologically sound?

A Registered Report may include valid exploratory analyses not previously outlined in the Registered Report Protocol, as long as they are described as such.

Reviewer #1: Yes

Reviewer #2: Yes

3. Are the conclusions supported by the data and do they address the research question presented in the Registered Report Protocol?

The manuscript must describe a technically sound piece of scientific research with data that supports the conclusions. The conclusions must be drawn appropriately based on the research question(s) outlined in the Registered Report Protocol and on the data presented.

Reviewer #1: Yes

Reviewer #2: No

4. Have the authors made all data underlying the findings in their manuscript fully available?

Reviewer #1: Yes

Reviewer #2: No

5. Is the manuscript presented in an intelligible fashion and written in standard English?

Reviewer #1: Yes

Reviewer #2: Yes

6. Review Comments to the Author

Please use the space provided to explain your answers to the questions above. (Please upload your review as an attachment if it exceeds 20,000 characters)

Reviewer #1: The authors responded to my comments on the previous round of revisions and I do not have any additional comments.

Reviewer #2: Authors have addressed most of my comments. I am still concerned about the following issues:

1. It is clear that Logistic model could overestimate effect size if prevalence is greater than 10% (a body of literature have addressed it). Generally this model is for issue of rare event or lower prevalence of event of interest.

2. Selection of variable should be based on objective of study and sample size. There was fewer covariates considered in this study with samples size of more than 800 and the aim is to identify the potential variables. It is uncessary to select by uivariate analysis. Authors cited the reference which is just for clinical prediction modelling where logisically limited number of covariates is necessary for prediction.

7. PLOS authors have the option to publish the peer review history of their article (what does this mean?). If published, this will include your full peer review and any attached files.

Reviewer #1: No

Reviewer #2: No

---

## [Author Response · Author response to Decision Letter 3]

28 May 2023

Reviewer #1: The authors responded to my comments on the previous round of revisions and I do not have any additional comments.

Response: We really appreciate the time you have put in to review this manuscript for the second time. Your comments have gone a long way to strengthen this manuscript.

Reviewer #2: Authors have addressed most of my comments. I am still concerned about the following issues:

Response: We really appreciate the efforts you have put in to strengthen this manuscript and thank you for sending back these comments. We hope that our answers meet your expectations. 

1. It is clear that Logistic model could overestimate effect size if prevalence is greater than 10% (a body of literature have addressed it). Generally this model is for issue of rare event or lower prevalence of event of interest.

 Response: We agree that logistic regression can overestimates effect size when the prevalence of the outcome is high. However, we feel that using logistic regression to bring out the information this study was designed for is not a problem. This is for the following reasons.

• Logistic regression and odds ratio is the most valid, if not, the only valid statistical method which is adapted for a cross sectional population study like ours. This method of analysis has been used in uncountable studies with relatively high prevalence of the outcome of interest [1–12].

• We feel that the error comes when the odds ratio is confused with the relative risk or prevalence ratio or directly translated to risk. When the results are reported in terms of increasing or reducing odds, the reader understands that in terms of magnitude of risk, it is not the same magnitude. In this particular study we did not intend to quantify the risk. Increasing or decreasing odds also mean increasing or decreasing risk even though not of the same magnitude [13].

• Logistic regression overestimates odds ratios in studies with small and moderate sample sizes [14]. This is not the case of our study which included more than 1000 women.

• The problem of the exact effect size or magnitude of risk is very important only in clinical trials where the magnitude of the odds ratio could directly be used to take decisions. Population studies evaluating behavior-linked practices identify more of factors and their direction of influence which is easy to state from our manuscript. Therefore, we donot really see the statistical method here affecting the outcome of this study.

• All reported results have been stated in the manuscript in terms of increasing and decreasing number of odds. 

2. Selection of variable should be based on objective of study and sample size. There was fewer covariates considered in this study with samples size of more than 800 and the aim is to identify the potential variables. It is uncessary to select by uivariate analysis. Authors cited the reference which is just for clinical prediction modelling where logisically limited number of covariates is necessary for prediction.

Response: Thanks very much for your worries. We belief this is with the objective of strengthening the work. We understand your worries. However, our covariates for potential risk factors were selected based on literature and its plausible potential to have an effect. All variables we felt could intervene were included in the logistic regression model. Some explanations to this had been included in the past reviews in the data analysis section page 11. What we don’t understand is whether there is a maximum number of variables we must have collected data on to be able to apply our model. We are surprised to have this as a problem because from publications we read every day, emphasis is not put on any limit of variables to reach.

We also acknowledged in the last paragraph of the discussion (page 24) that we did not collect data on all potential factors which could influence calcium supplementation adherence as a strong limit of our study. 

References

1. Tesfaye A, Alemayehu M, Abere G, Mekonnen T. Prevalence and Associated Factors of Computer Vision Syndrome Among Academic Staff in the University of Gondar, Northwest Ethiopia: An Institution-Based Cross-Sectional Study. Environ Health Insights. 2022;16: 117863022211118. doi:10.1177/11786302221111865

2. Tesfaye A, Mekonnen T, Alemayehu M, Abere G. Prevalence and Risk Factors of Work-Related Upper Extremity Disorders among University Teaching Staff in Ethiopia, 2021: An Institution-Based Cross-Sectional Study. Pain Res Manag. 2022;2022: 1–12. doi:10.1155/2022/7744879

3. Tang L, Lee AH, Yau KKW, Hui YV, Binns CW. Consumption of dietary supplements by Chinese women during pregnancy and postpartum: A prospective cohort study. Matern Child Nutr. 2017. doi:10.1111/mcn.12435

4. He G, Yang H, Chen M, Liu X. Factors influencing folic acid, multivitamin, and calcium supplementation among pregnant women in China based on a national cross-sectional survey. Chin Med J (Engl). 2023;136: 473. doi:10.1097/CM9.0000000000002202

5. Adimasu A, Azene Z, Merid M, Goshu A, Geberu D, Kassa G, et al. Knowledge and attitude of the communities towards COVID-19 and associated factors among Gondar City residents, northwest Ethiopia: A community based cross-sectional study. PLOS ONE. 2021;16: e0248821. doi:10.1371/journal.pone.0248821

6. Wami S, Mekonnen T, Yirdaw G, Abere G. Musculoskeletal problems and associated risk factors among health science students in Ethiopia: a cross-sectional study. J Public Health. 2021;29: 1–7. doi:10.1007/s10389-020-01201-6

7. Mekonnen T, Tesfaye Y, Moges H, Berhe R. Factors associated with risky driving behaviors for road traffic crashes among professional car drivers in Bahirdar city, northwest Ethiopia, 2016: A cross-sectional study. Environ Health Prev Med. 2019;24. doi:10.1186/s12199-019-0772-1

8. Geberu D, Biks G, Gebremedhin T, Mekonnen T. Factors of patient satisfaction in adult outpatient departments of private wing and regular services in public hospitals of Addis Ababa, Ethiopia: a comparative cross-sectional study. BMC Health Serv Res. 2019;19. doi:10.1186/s12913-019-4685-x

9. Mekonnen T. The magnitude and factors associated with work-related back and lower extremity musculoskeletal disorders among barbers in Gondar town, northwest Ethiopia, 2017: A cross-sectional study. PLOS ONE. 2019;14. doi:10.1371/journal.pone.0220035

10. Utami A, Margawati A, Pramono D, Nugraheni A, Pramudo SG. Determinant Factors of COVID-19 Vaccine Hesitancy Among Adult and Elderly Population in Central Java, Indonesia. Patient Prefer Adherence. 2022;16: 1559–1570. doi:10.2147/PPA.S365663

11. Berihun G, Walle Z, Teshome D, Berhanu L, Derso M. COVID-19 Vaccine Acceptance and Associated Factors Among College Students in Dessie City, Northeastern Ethiopia. J Multidiscip Healthc. 2022;15: 1735–1746. doi:10.2147/JMDH.S381151

12. Gedif G, Sisay Y, Alebel A, Belay YA. Level of job satisfaction and associated factors among health care professionals working at University of Gondar Referral Hospital, Northwest Ethiopia: a cross-sectional study. BMC Res Notes. 2018;11: 824. doi:10.1186/s13104-018-3918-0

13. Barros AJ, Hirakata VN. Alternatives for logistic regression in cross-sectional studies: an empirical comparison of models that directly estimate the prevalence ratio. BMC Med Res Methodol. 2003;3: 21. doi:10.1186/1471-2288-3-21

14. Nemes S, Jonasson JM, Genell A, Steineck G. Bias in odds ratios by logistic regression modelling and sample size. BMC Med Res Methodol. 2009;9: 56. doi:10.1186/1471-2288-9-56

---

## [Decision Letter · Decision Letter 4]

18 Sep 2023

Calcium supplementation in pregnancy:  an analysis of potential determinants in an under-resourced setting

PONE-D-21-31679R4

Dear Dr. Ajong,

We’re pleased to inform you that your manuscript has been judged scientifically suitable for publication and will be formally accepted for publication once it meets all outstanding technical requirements.

Kind regards,

Avanti Dey, PhD

Staff Editor

PLOS ONE

Additional Editor Comments (optional):

Reviewers' comments:

Reviewer's Responses to Questions

**Comments to the Author**

1. Does the manuscript adhere to the experimental procedures and analyses described in the Registered Report Protocol?

If the manuscript reports any deviations from the planned experimental procedures and analyses, those must be reasonable and adequately justified.

Reviewer #1: Yes

2. If the manuscript reports exploratory analyses or experimental procedures not outlined in the original Registered Report Protocol, are these reasonable, justified and methodologically sound?

A Registered Report may include valid exploratory analyses not previously outlined in the Registered Report Protocol, as long as they are described as such.

Reviewer #1: Yes

3. Are the conclusions supported by the data and do they address the research question presented in the Registered Report Protocol?

The manuscript must describe a technically sound piece of scientific research with data that supports the conclusions. The conclusions must be drawn appropriately based on the research question(s) outlined in the Registered Report Protocol and on the data presented.

Reviewer #1: Yes

4. Have the authors made all data underlying the findings in their manuscript fully available?

Reviewer #1: Yes

5. Is the manuscript presented in an intelligible fashion and written in standard English?

Reviewer #1: Yes

6. Review Comments to the Author

Please use the space provided to explain your answers to the questions above. (Please upload your review as an attachment if it exceeds 20,000 characters)

Reviewer #1: The authors had addressed my comments in my previous review. An issue I noted in this read is the reference to the survey as a semi-structured questionnaire. In looking at the survey instrument in the protocol paper it seems like this was not semi-structured and likely was followed exactly as written. I suggest referring to it as survey interviews or structured questionnaire or something other than semi-structured, unless of course it was semi-structured.

7. PLOS authors have the option to publish the peer review history of their article (what does this mean?). If published, this will include your full peer review and any attached files.

Reviewer #1: No

---

## [Editor Report · Acceptance letter]

28 Sep 2023

PONE-D-21-31679R4 

Calcium supplementation in pregnancy:  an analysis of potential determinants in an under-resourced setting 

Dear Dr. Ajong:

I'm pleased to inform you that your manuscript has been deemed suitable for publication in PLOS ONE. Congratulations! Your manuscript is now with our production department. 

Kind regards, 

on behalf of

Dr. Avanti Dey 

Staff Editor

PLOS ONE